

# Submicron aerosol composition in the world's most polluted megacity: The Delhi Aerosol Supersite campaign

Shahzad Gani[1], Sahil Bhandari[2], Sarah Seraj[1], Dongyu S. Wang[2], Kanan Patel[2], Prashant Soni[3], Zainab Arub[3], Gazala Habib[3], Lea Hildebrandt Ruiz[2], and Joshua S. Apte[1]

[1]Department of Civil, Architectural and Environmental Engineering, The University of Texas at Austin, Texas, USA
[2]McKetta Department of Chemical Engineering, The University of Texas at Austin, Texas, USA
[3]Department of Civil Engineering, Indian Institute of Technology Delhi, New Delhi, India

**Correspondence:** Joshua S. Apte (jsapte@utexas.edu), Lea Hildebrandt Ruiz (lhr@che.utexas.edu)

**Abstract.**

Delhi, India routinely experiences some of the world's highest urban particulate matter concentrations. We established the Delhi Aerosol Supersite campaign to provide long-term characterization of the ambient submicron aerosol composition in Delhi. Here we report on 1.25 years of highly time resolved speciated submicron particulate matter ($PM_1$) data, including

black carbon (BC) and non-refractory $PM_1$ (NR-$PM_1$), which we combine to develop a composition-based estimate of $PM_1$ ("C-$PM_1$" = BC + NR-$PM_1$) concentrations.

We observed marked seasonal and diurnal variability in the concentration and composition of $PM_1$ owing to the interactions of sources and atmospheric processes. Winter was the most polluted period of the year with average C-$PM_1$ mass concentrations of ~210 µg m$^{-3}$. Monsoon was hot and rainy, consequently making it the least polluted (C-$PM_1$ ~50 µg m$^{-3}$) period.

Organics constituted more than half of the C-$PM_1$ for all seasons and times of day. While ammonium, chloride and nitrate each were ~10% of the C-$PM_1$ for the cooler months, BC and sulfate contributed ~5% each. For the warmer periods, the fractional contribution of BC and sulfate to C-$PM_1$ increased and the chloride contribution decreased to less than 2%. The seasonal and diurnal variation in absolute mass loadings were generally consistent with changes in ventilation coefficients, with higher concentrations for periods with unfavorable meteorology—low planetary boundary layer height and low wind speeds. However,

the variation in C-$PM_1$ composition was influenced by temporally varying sources, photochemistry and gas-particle partitioning. During cool periods when wind was from the northwest, episodic hourly averaged chloride concentrations reached 50–100 µg m$^{-3}$, ranking among the highest chloride concentrations reported anywhere in the world.

We estimated the contribution of primary emissions and secondary processes to Delhi's submicron aerosol. Secondary species contributed almost 50–70% of Delhi's C-$PM_1$ mass for the winter and spring months, and up to 60–80% for the warmer

summer and monsoon months. For the cooler months that had the highest C-$PM_1$ concentrations, the nighttime sources were skewed towards primary sources, while the daytime C-$PM_1$ was dominated by secondary species. Overall, these findings point to the important effects of both primary emissions and more regional atmospheric chemistry on influencing the extreme particle concentrations that impact the Delhi megacity region. Future air quality strategies considering Delhi's situation in both a regional and local context will be more effective than policies targeting only local, primary air pollutants.





## 1 Introduction

Outdoor air pollution has detrimental health effects (Pope and Dockery, 2006) and is responsible for more than 4 million deaths every year globally (Cohen et al., 2017), resulting in substantial global and regional decrements in life expectancy (Apte et al., 2018). India experiences high ambient air pollution with an annual population weighted $PM_{2.5}$ (particulate matter with

diameter less than 2.5 µm) mean of 74 µg m$^{-3}$ and experiences the highest number of deaths from ambient air pollution among all countries in the world (~1.1 million people yr$^{-1}$, ~1.5 yr of life lost due to air pollution) (Cohen et al., 2017; Apte et al., 2018). Some of the most polluted cities in the world are in India. Delhi (population = 28 million) is the world's most polluted megacity, with recent annual-average $PM_{2.5}$ concentrations of ~140 µg m$^{-3}$ (World Health Organization, 2018).

Previous aerosol characterization campaigns in Delhi have noted the importance of both primary and secondary sources

to Delhi's poor ambient air quality (Jaiprakash et al., 2017; Pant et al., 2015). These studies have shown Delhi's PM to be rich in organics throughout the year and to contain high concentrations of inorganic species such as chloride and nitrate during the foggy wintertime. Furthermore, high concentrations of black carbon (BC) and brown carbon attributable to primary emissions such as biomass combustion and diesel exhaust have been observed across North India (Gupta et al., 2017; Satish et al., 2017; Bhat et al., 2017). However, previous studies in Delhi have mostly observed aerosol composition for short periods

with limited temporal information (Pant et al., 2016). The Delhi Aerosol Supersite (DAS) campaign was designed to address current uncertainties in Delhi's aerosol composition by collecting continuous, highly time-resolved data on a long-term basis. In addition to providing insights into the atmospheric processes relevant for a polluted megacity, this study contributes to the understanding of the atmospheric science for South Asia in general. The lessons from Delhi have relevance for the entire Indo-Gangetic plain (population: ~400 million; including parts of India, Pakistan, Bangladesh and Nepal) that experiences similar

meteorology and high PM levels, especially during wintertime (Kumar et al., 2017; Singh et al., 2015).

Here we provide a detailed overview of the chemical composition of $PM_1$ in Delhi by season and time of day based on a long-term deployment of a mass spectrometer instrument. We also provide insights into the role of meteorology on the concentration and composition of $PM_1$. Finally, we include a brief overview of the source apportionment results from the positive matrix factorization (PMF) of aerosol mass spectra to understand the contribution of primary emissions and secondary

processes to Delhi's PM concentrations, with details of the PMF provided in a companion paper (Bhandari et al., 2018).

## 2 Methods

### 2.1 Sampling site and pollutants measured

Delhi experiences a wide range of variation in temperature (T), relative humidity (RH), wind speeds, and precipitation across the year and by time of day (Fig. 1). The winters (December to mid-February) are cool (T ~10–20°C, average diurnal range)

and humid (RH ~45–90%) with low wind speeds (~2–3 ms$^{-1}$). Delhi frequently experiences shallow inversion layers (depth < 100 m) during the winter, especially at night and in the morning hours. Summers (April to June) are very hot (T ~25–40°C) and dry (RH ~30–55%). Delhi and most of the Indo-Gangetic plain experiences episodic heavy rainfall during the monsoon (July



to mid-September), accompanied by slightly lower temperatures (T ~25–35°C) than the summers. While the winds throughout most of the year are predominately from the northwest, during the monsoon the wind direction are from the south during the nighttime. Spring (mid-February to March) and autumn (mid-September to November) are periods of transition between these meteorological extremes. For all seasons, the ventilation coefficient is highest during the daytime when the boundary layer

height and the wind speeds reach their diurnal maxima. Changes in ventilation play an important role in the large seasonal and diurnal variation of PM (Trivedi et al., 2014). Unfavorable meteorological conditions often amplify primary emissions to produce spectacularly high $PM_{2.5}$ concentrations (Guttikunda and Gurjar, 2012).

To investigate the composition of ambient air in New Delhi at high time resolution, we installed a suite of online aerosol measurement instrumentation at the Indian Institute of Technology Delhi (IITD) campus in South Delhi. The instruments are

situated in a temperature-controlled laboratory on the top floor of a 4-story building. The nearest source of local emissions is an arterial road located 150 m away from the building. We measured chemical composition of non-refractory $PM_1$ (NR-$PM_1$) using an Aerodyne Aerosol Chemical Speciation Monitor (ACSM, Aerodyne Research, Billerica MA). BC was measured using a multi-channel aethalometer (Magee Scientific Model AE33, Berkeley, CA) with a multi-spot sampling system designed to minimize the filter loading artifact present on earlier aethalometer systems (Drinovec et al., 2015). Particle size distributions

(PSD) were measured using a scanning mobility particle sizer (SMPS, TSI, Shoreview, MN) consisting of an electrostatic classifier (TSI model 3080), differential mobility analyzer (DMA, TSI model 3081), X-ray aerosol neutralizer (TSI model 3088), and a water-based condensation particle counter (CPC, TSI model 3785).

## 2.2   Instrumentation

The instruments were placed on two separate sampling lines. The first sampling line ($SL_1$) had the ACSM and the SMPS in

parallel. The second sampling line ($SL_2$) was for the aethalometer. Both sampling lines had a $PM_1$ cyclone at the inlet, followed by a water trap and a Nafion membrane diffusion dryer (Magee Scientific sample stream dryer, Berkeley, CA). The flow rate in $SL_1$ was 3 LPM, divided as follows: 1 LPM pulled by the SMPS, 0.1 LPM by the ACSM, and the remaining 1.9 LPM by an in-line flow controller which was in parallel with the SMPS and downstream from the ACSM. $SL_2$ had a flow rate of 2 LPM pulled by the aethalometer. For the SMPS, the CPC pulled at a 1 LPM flow rate and the electrostatic classifier was operated

at a sheath flow rate of 4 LPM to enable SMPS scanning over a broad range of particle sizes. We conducted experiments at multiple sheath flow rates from 4 to 10 LPM and found the results to be consistent.

The ACSM measures NR-$PM_1$, i.e., those compounds that flash vaporize at the heater temperature of ~600°C. The flash-vaporized compounds are subsequently ionized in the ACSM via 70eV electron impact ionization and detected with a quadrupole mass spectrometer (Ng et al., 2011c). The scan speed was set at 200 ms amu$^{-1}$ and pause setting at 125 for a sampling time (64

seconds). Detailed operational procedures for the ACSM are provided in Appendix A1. Some submicron aerosol constituents are refractory, including BC, metals, and crustal materials. For our core analyses of $PM_1$ mass, we use the sum of NR-$PM_1$+BC as a composition-based proxy for total $PM_1$, which we term "C-$PM_1$". This C-$PM_1$ metric excludes the contribution to submicron mass of refractory metals and crustal materials, which we estimate results in a 5–10% underestimate of total $PM_1$ mass (see below).



## 2.3 Data processing

The SMPS scanned from 12 to 560 nm with each subsequent scan 135 seconds apart. We used a mode fitting algorithm (Hussein et al., 2005) in the mass domain to estimate the PSD between 560 and 1000 nm. We validated the performance of our model by comparing the modeled and observed volume and number concentrations for the observed particle size range. We found that

the model predicted the same volume as was observed (slope = 1.00, $R^2$ = 1.00), but slightly overestimated particle number concentrations (slope = 1.06, $R^2$ = 0.96), mostly for smaller particles. In order to develop a supplemental PSD-based estimate of submicron mass, we first estimated a complete (hybrid) PSD up to 1000 nm by combining the observed PSD from 12 to 560 nm and the modeled PSD from 560 to 1000 nm. Estimates of aerosol densities from Asia range between 1.3–1.6 g cm³ (Sarangi et al., 2016; Hu et al., 2012). Using a particle density of 1.6 g cm⁻³ and the hybrid PSD, we developed a SMPS-based $PM_1$

estimate ("SMPS-$PM_1$"). On an hourly basis, the linear fit between our core C-$PM_1$ and supplemental SMPS-$PM_1$ estimates had a slope of 0.96 and an $R^2$ of 0.85 (Fig. S1). This linear fit suggested that our speciated $PM_1$ data (NR-$PM_1$ species and BC) agreed reasonably well with the SMPS-$PM_1$ estimates. We used the PSD to estimate the transmission efficiency (TE) of the ACSM. The details of this correction along with other ACSM data processing steps are provided in Appendix A2. We estimate an overall uncertainty of up to 20–25% in the ACSM data, which is within expectations for measurements from this instrument

(Crenn et al., 2015).

While we acquired data for each instrument at high time resolution (~1 minute for aethalometer and ACSM; ~2 minutes for SMPS), for analytical simplicity we generally present the hourly averaged data for each instrument in this study. We categorize the seasons as winters (December to mid-February), summers (April to June), monsoon (July to mid-September) and spring (mid-February to March) (Indian National Science Academy, 2018). Autumn (mid-September to November) is not included in

our core analyses due to the unavailability of ACSM data for that period.

We retrieved the hourly temperature and relative humidity (RH) data from the Indira Gandhi International Airport (IGIA, 8 km from our site). To obtain mesoscale (regional) meteorological data for wind speed, direction (10m from ground) and planetary boundary layer height (PBLH) in Delhi, we used a NASA meteorological reanalysis dataset, MERRA2 (Gelaro et al., 2017). MERRA2 has a spatial resolution of 0.5° × 0.625° (55 km × 60 km) and an hourly temporal resolution. We

retrieved daily precipitation data for Delhi from the European Centre for Medium-Range Weather Forecasts' reanalysis dataset, ERA-Interim (Dee et al., 2011).

The hourly data for all species across the campaign are neither strictly normally nor log-normally distributed (Fig. S2). However, since the data are relatively closer to being log-normally distributed, we have included geometric mean (GM) and geometric standard deviation (GSD) in addition to the arithmetic mean (AM) wherever possible to provide a more complete

representation of the central tendency of the data. Furthermore, the annual averages reported in this study are the averages of all the available hourly data from 2017 for the NR-$PM_1$ species and BC. It should be noted that we do not have ACSM data (NR-$PM_1$ species) for autumn and only few days of aethalometer data (BC) for monsoon. On the basis of available SMPS-$PM_1$ data (our site) and $PM_{2.5}$ data (multiple regulatory monitors in Delhi), we estimate that the true annual average differs from the data we collected by within ±20%. As a sensitivity analysis, we reconstruct annual and campaign averages by giving equal



weight to each 2-month period. For example, to calculate the synthetic (reconstructed) annual average for 2017, we averaged the averages of the six 2-month periods (January-February through November-December). In Table S1 we have provided a comparison between the AM, GM and the synthetic averages of the $PM_1$ components for the 2017 data against the entire campaign data.

## 3 Results and discussion

### 3.1 Mass concentration

We observed marked seasonal and diurnal variation in the PM mass concentration owing to the interactions of sources, atmospheric mixing, and physicochemical processing. Figure 2 shows the time series of NR-$PM_1$, individual submicron species, $PM_{2.5}$ at a background site, and selected meteorological parameters. The daily average NR-$PM_1$ concentrations at our site varied between 12.7 and 392 µg m$^{-3}$ with an annual average of 87.3 µg m$^{-3}$. Most C-$PM_1$ mass was nonrefractory—the average NR-$PM_1$ fraction of C-$PM_1$ was highest in the winter (94%) and lowest for warmer months (85%) (Fig. 3). The average wintertime NR-$PM_1$ concentration was ~2× higher than spring and ~4× higher than the warmer months. Using speciated mass concentrations and the PSD, we observed that C-$PM_1$ was highly correlated with SMPS-$PM_1$ ($R^2$ = 0.83) and we achieved almost complete mass closure (Fig. S1). That most of the $PM_1$ was composed of nonrefractory material and BC was consistent with past literature from Delhi which observed that metals and other nonrefractory crustal materials, which we did not measure in this study, constituted less than 5% of $PM_1$ (Jaiprakash et al., 2017).

We estimated that the C-$PM_1$ concentrations observed at our site were generally ~85% of the $PM_{2.5}$ concentrations ($R^2$ = 0.54 and slope = 0.85 for linear fit of hourly C-$PM_1$ and $PM_{2.5}$ concentrations over entire campaign) measured at the nearest monitoring station that is operated by the Delhi Pollution Control Committee (DPCC), R.K. Puram (3 km away), where the annual average $PM_{2.5}$ concentration for 2017 was 140 µg m$^{-3}$. There were strong seasonal and diurnal variations in mass loadings with winter being the most polluted with average concentration ~4× higher than the least polluted summer and monsoon months. The daily average $PM_{2.5}$ concentration exceeded the daily average Indian National Ambient Air Quality Standards (NAAQS, 60 µg m$^{-3}$) on more than 80% of the days and the World Health Organization (WHO) 24-hour average air quality guidelines (25 µg m$^{-3}$) on all but two days. A distinct feature of Delhi's wintertime air pollution is the nearly complete absence of periods of clean air, in contrast to some other polluted megacities (e.g., Beijing), which are characterized by episodic alternation between clean and polluted conditions (Sun et al., 2013). For wintertime, the daily average $PM_{2.5}$ concentrations exceeded 100 µg m$^{-3}$ on 94% of the days, the Indian NAAQS on 99% of the days, and the WHO guidelines on all days. Daily-average $PM_{2.5}$ concentrations at R.K. Puram exceeded 500 µg m$^{-3}$ on four days in 2017.

### 3.2 $PM_1$ composition: seasonal and diurnal variation

The concentrations and fractional contribution to $PM_1$ of each species varied by season and time of day. Over the campaign, organics comprised of 54% of the submicron mass, inorganics (chloride, ammonium, nitrate and sulfate) 36% and BC 10%.





There was a strong seasonality in C-PM$_1$ loadings with the wintertime average loadings exceeding the relatively less polluted and warmer summer and monsoon months by 3–4×. We report the average seasonal concentrations of organics, sulfate, ammonium, nitrate, chloride and BC in Table 1 and their contribution to C-PM$_1$ in Fig. 3. Within each season there were distinct diurnal (time-of-day) trends for the average concentrations by hour of day for NR-PM$_1$ and PM$_1$ components (Fig. 4). These

diurnal swings of the average hourly concentrations were the most prominent for the colder winter and spring months. In winter, average hourly NR-PM$_1$ concentrations ranged between 97.4 and 254 µg m$^{-3}$ (minimum and maximum concentrations for the average diurnal cycle). Spring conditions were moderately less polluted, with hourly average concentrations ranging diurnally from 37.0 to 167 µg m$^{-3}$. The NR-PM$_1$ concentrations varied much less during summer (range of concentrations for an average diurnal cycle: 38.7 to 72.4 µg m$^{-3}$) and monsoon (32.1 to 47.7 µg m$^{-3}$). For most seasons, the hourly averaged

NR-PM$_1$ concentrations peaked around 7–8 AM and then again around 9–10 PM at night, with the daily minimum typically occurring around 3–4 PM. However, for the monsoon months the NR-PM$_1$ average hourly concentrations were similar throughout the day. The diurnal variation in average hourly concentrations and fractional composition of NR-PM$_1$ species for each season is presented in Fig. 5. The day and night averages by season for each PM$_1$ species along with the summary averages of meteorological parameters are presented in Table 2. We did not observe any marked day of week difference in the levels or

composition of C-PM$_1$ (Fig. S3).

Organics were the single largest C-PM$_1$ mass component for all seasons and at all times of the average diurnal cycle. Organics consistently contributed to more than ~50% of seasonal C-PM$_1$ mass, with some episodes when their contribution was as high as 80%. The high organic fraction of PM is consistent with studies from across the world (Zhang et al., 2007; Jimenez et al., 2009). The daily averages of organics at our site varied between 6.4 and 293 µg m$^{-3}$ with an annual average of 51.5 µg m$^{-3}$.

The average wintertime organic concentration was ~2× higher than spring and ~4–5× than summer and monsoon. While the wintertime organic concentrations ranged between 53.3 and 166 µg m$^{-3}$ with the highest concentration during the night, the diurnal variations were less dynamic for the warmer months with the hourly average organic concentrations ranging between 20.8 and 49.8 µg m$^{-3}$ for summertime. Comparing daytime and nighttime $f_{43}$ and $f_{44}$ values for each season, the bulk organic aerosol was generally more oxidized during the warmer periods (Fig. S4), presumably owing to the higher photochemical

activity during that time (Ng et al., 2011a).

Ammonium was the prominent inorganic cation in C-PM$_1$ and generally balanced all the anionic inorganic species (chloride, nitrate and sulfate). Over the campaign, the molar ratio of the inorganic anions to cations (ammonium) was 0.98 ($R^2 = 0.89$), with sufficient ammonium concentrations to neutralize the anionic species present for ~85% of the hourly data. Ammonium mass concentrations were consistently around ~10% of the observed C-PM$_1$. The daily average of ammonium at our site varied

between 1.5 and 37.9 µg m$^{-3}$ with an annual average of 9.0 µg m$^{-3}$. The average ammonium concentration for wintertime was ~2× higher than spring and ~4× higher than summer and monsoon. Ammonium concentration hourly averages ranged between 10.9 to 30.8 µg m$^{-3}$ during winters and 4.2 to 8.3 µg m$^{-3}$ during the summer.

We observed some of the highest chloride concentrations reported anywhere in the world with episodes when hourly averages exceeded 100 µg m$^{-3}$ (more than 40 such hours across the campaign). The 90[th] and 95[th] percentile values of the hourly

concentrations of chloride over the campaign were 26.7 and 43.8 µg m$^{-3}$ respectively. The daily average of chloride concen-



tration at our site varied between 0.1 and 66.6 µg m$^{-3}$, with an annual average of 6.1 µg m$^{-3}$. Chloride concentrations showed the strongest seasonal variability with the average wintertime concentration ~2× higher than those during spring and more than 20× higher than those during the warm (summer and monsoon) months. During the cooler winter and spring months, chloride concentrations had the largest diurnal variation among all PM species observed, with the average diurnal minimum and max-

imum hourly concentration ranging between 4.6 and 47.3 µg m$^{-3}$ in wintertime. The winter chloride peak is notable for its timing in the early morning hours (~7 AM), which is considerably later than the diurnal peak in organics and BC, which tended to occur shortly before midnight. While chloride contributed more than 10% of the submicron mass in the winters, it comprised of less than 2% of the C-PM$_1$ mass concentration during the summer and monsoon. Furthermore, chloride constituted around 12–16% of NR-PM$_1$ for temperatures below 15°C, but dropped to less than 4% of the NR-PM$_1$ concentrations for temperatures

above 25°C (Fig. S5). Given that ammonium was nearly always present in sufficient quantities to neutralize the major inorganic anions measured by the ACSM, we infer that the dominant fraction of chloride was usually present as ammonium chloride, for which gas-particle partitioning is strongly temperature dependent. (However, we cannot exclude the possibility that organic chlorides contributed a subsidiary fraction of the chloride mass.) Even for high episodic chloride concentrations, ammonium was present in sufficient levels to nearly neutralize the anionic species, with a deficiency of only 12% when considering only

hours with chloride concentrations higher than the 90$^{th}$ percentile campaign value (26.7 µg m$^{-3}$).

To understand whether the sharp drop in chloride concentrations for warmer times of day could be explained by evaporation of ammonium chloride, we used an inorganic aerosol thermodynamics model (Friese and Ebel, 2010). The detailed methodology and results of the model used are presented in Appendix B. Results from inorganic thermodynamic modeling suggest that all ammonium chloride observed in the winter is expected to evaporate at summer temperatures and relative humidity,

consistent with our observations. The volatile nature of ammonium chloride has also been observed in other parts of the world (Salcedo et al., 2006; Wang et al., 2016) and was consistent with the sharp drop in the chloride fraction that we observed for the warmer periods. We believe that gas-particle partitioning and episodic sources (Sec. 3.4.1) may drive much of the diurnal and seasonal variation in particulate chloride. We would therefore expect a large fraction of chloride to be in the gas-phase, especially for warmer periods. We did not collect gas-phase HCl measurements here, but future studies could validate this

hypothesis through measurements of gas-phase chloride.

Nitrate comprised 6–12% of Delhi's C-PM$_1$ with daily averages between 0.6 and 58.5 µg m$^{-3}$, with an annual average of 8.8 µg m$^{-3}$. The average wintertime nitrate concentration was more than 2× higher than spring and more than 6× higher than summer and monsoon. The average diurnal cycle (lowest and highest hourly average concentration) for wintertime concentrations ranged between 15.9 and 33.6 µg m$^{-3}$ and the summer concentrations ranged between 2.2 and 6.3 µg m$^{-3}$. The nitrate fraction

of NR-PM$_1$ dropped from 12% at temperatures below 25°C to 5–9% at temperatures above 25°C, likely due to the thermodynamics of ammonium nitrate. As with chloride, nitrate concentrations were also generally highly correlated with ammonium concentrations ($R^2$ = 0.69 for hourly data over entire campaign), suggesting that most of the nitrate observed was present as ammonium nitrate. Considering the ubiquitous NO$_x$ sources in this megacity, organic nitrates may also contribute to the total nitrate measured by the ACSM.





The daily averages of sulfate at our sites varied between 3.1 and 34.5 μg m$^{-3}$ with an annual average of 11.8 μg m$^{-3}$. Sulfate had the least seasonal variability among the NR-PM$_1$ species with wintertime average concentration ~1.5× higher than each of the other seasons (spring, summer and monsoon). In addition to the low seasonal variability, sulfate was also the chemical constituent with the least diurnal variation and had relatively higher daytime concentrations for the warmer summer months.

The diurnal variation in sulfate concentration for the cooler months was similar to those of other PM$_1$ species with the average early morning concentrations for winter and spring almost 2× higher than the daytime concentrations. Sulfate was the only NR-PM$_1$ species that had a higher mass fraction during the warmer months, contributing 13–30% to the C-PM$_1$ mass in the warmer months, 8–20% for spring and 5–13% for winter, with the mass fraction being highest during the daytime for all seasons. The sulfate fraction of NR-PM$_1$ increased from less than 10% for periods cooler than 25°C to more than 25% for periods above

35°C. The increase in sulfate mass fraction for warmer periods can be explained by the lower diurnal and seasonal variability in its absolute concentration, possibly due to a combination of increased daytime photochemical formation rates for warmer months and sulfate being well mixed in the atmosphere because of its transport from longer distances (Verma et al., 2012).

BC contributed to 6.4% of the C-PM$_1$ mass concentration in the winter compared to 10% in the spring, and 14% in the summer. We had limited monsoon data for BC. The daily average of BC at our site varied between 2.2 and 35.2 μg m$^{-3}$

with an annual average of 12.4 μg m$^{-3}$. The average wintertime BC concentration was ~1.5× higher than spring and summer. The seasonal differences in the absolute BC concentrations were not as high as any of the other PM$_1$ species. One possible explanation for this result relates to the presence of nearby BC sources within Delhi, including the major ring road with truck traffic near our sampling site. These trucks are often restricted to pass through Delhi only at night (Guttikunda and Calori, 2013). It is plausible that these nearby primary emissions would be incompletely mixed into the boundary layer, and are

therefore relatively less affected by atmospheric mixing (Sec. 3.3 and Sec. 3.5). Accordingly, BC had sharp diurnal variability, with peak nocturnal BC concentrations typically ~3-4× higher than during mid-day hours, with peak concentrations occurring at a similar time to the temporal peak for organics (typically just before midnight).

## 3.3    Role of meteorology

The planetary boundary layer height (PBLH) had a strong seasonal variation with summer heights 2–4× larger than those during

the cooler months. The seasonal variability in the PBLH along with those in wind speeds resulted in the ventilation coefficient (VC = PBLH × wind speed, sometimes referred to as normalized dilution rate) 4–6× slower for the wintertime compared to the summer. VC is often used as a parameter to characterize the role of atmospheric dilution to pollutant concentrations both in the Indian context (Vittal Murty et al., 1980) and globally (Marshall et al., 2005; Apte et al., 2012). Seasonal variability in VC appear to reasonably agree with the higher NR-PM$_1$ concentrations in less ventilated cold months and lower concentrations

in the warmer months when VC was higher (Fig. 6). The week with the lowest VC was ~6× less ventilated than the most ventilated week, and had ~6× higher NR-PM$_1$ mass concentrations. For the non-monsoon periods, the VC was generally a good indicator of NR-PM$_1$ concentrations (R$^2$ = 0.56 for weekly averaged data). For the cooler winter and spring months, the R$^2$ for the linear fit of the weekly averaged VC and NR-PM$_1$ concentrations was 0.79. The monsoon concentrations were lower than those that would be expected by the VC calculated for those periods. This result could be explained by a combination of





change in the prominent nighttime wind direction from northwest to south during the monsoon and the washout of PM by the monsoon rain. For the monsoon period, we observed that the average NR-PM$_1$ mass concentration was almost half on days when it rained compared to the dry (no rain) days with no change in the composition of NR-PM$_1$. The strong modulating effect of meteorology on air pollution is well appreciated for Delhi and other Indian cities (Guttikunda and Gurjar, 2012; Tiwari et al.,

2015; Sujatha et al., 2016).

Even within each season, the VC showed large time of day variations with highest hourly average values 5–10× larger than the lowest. For each season, times of day with lower VC had the highest NR-PM$_1$ concentrations and the concentrations decreased as the VC value increased (Fig. S7). The large diurnal range of VC seemed to explain most of the variability in NR-PM$_1$ concentrations by time-of-day for most seasons ($R^2_{winter}$ = 0.88; $R^2_{spring}$ = 0.93; $R^2_{summer}$ = 0.81). For monsoon, the

diurnal variability of most PM$_1$ species was generally low even though VC varied by time-of-day, possibly due to precipitation washout of PM and change in characteristic wind direction during monsoon (as discussed above).

In general, the sharp variation in VC by season and time of day appear to explain much of the variability in NR-PM$_1$ concentrations. Furthermore, volatile species (e.g., ammonium chloride and ammonium nitrate) evaporate to the gas phase during warmer periods, further lowering the mass concentrations compared to the cooler periods. While there are seasonal

differences in emissions from sources such as crop burning and local biomass burning for heat (Guttikunda and Calori, 2013), our analysis perhaps suggests that in addition to episodic sources, meteorology being unfavorable is an important factor for some of the high PM concentrations observed.

### 3.4  Episodic high concentrations

#### 3.4.1  Chloride episodes and wind direction

Delhi experiences a prominently northwestern wind (Fig. 1). However, we observed that for brief periods during winter and spring when the wind direction was from the south, the peak chloride concentrations dropped from as high as 50–100 μg m$^{-3}$ on one day to less than 10 μg m$^{-3}$ on the next (Fig. 7). Furthermore, the highest decile of chloride concentrations in the campaign were mostly observed when the wind was from the northwest (Fig. 8). During winter mornings, when chloride concentrations were generally highest, the chloride fraction of C-PM$_1$ was almost 2× higher for periods with a northwestern wind compared

to periods with wind from any other direction (Fig. S8). These findings suggest a large source of chloride in the northwest of Delhi. The high levels of chloride observed in Delhi are neither observed in other South Asian countries (Kim et al., 2015; Stone et al., 2010; Salam et al., 2003), nor in other parts of India (Gupta and Mandariya, 2013; Guttikunda et al., 2013; Gupta et al., 2007), suggesting that these extreme levels of chloride probably come from more than just the usual type of biomass and waste burning (Goetz et al., 2018) which is ubiquitous across South Asia (Streets et al., 2003). Furthermore, the PMF factor for

biomass burning organic aerosol of Bhandari et al. (2018) does not correlate with chloride. There are many industrial sites in the northwest of Delhi, including metal processing plants that use HCl for steel pickling (Jaiprakash et al., 2017). The fugitive HCl fumes from these industries along with the high ammonia in Delhi (Warner et al., 2017) could be a pathway for these high particulate chloride concentrations observed (Pio and Harrison, 1987). Other possible sources of HCl are from the combustion



of polyvinyl chloride, coal, and biomass burning (Yudovich and Ketris, 2006; Lightowlers and Cape, 1988; Palmer, 1976). Our findings are based on measurement of particulate chloride and inorganic thermodynamic modeling, and can be tested by future studies that measure both gas and particulate chloride.

### 3.4.2 High organic episode

While organics contributed to almost half of the C-PM$_1$ for all seasons and times of day, there were episodes for which the contribution of organics increased to as high as 80% of the C-PM$_1$. One such episode was around Lohri (13th January, 2018), a festival celebrated in many parts of north India (including Delhi and regions up-wind of Delhi) with bonfires burnt at night. In 2018, Lohri was on a weekend (Saturday) and we observed a sharp increase with nighttime C-PM$_1$ concentrations almost 2–3× higher than the weekday nights preceding Lohri (Fig. 9). The contribution of both organics and BC increased for this period,

with organics concentrations as high as 300 μg m$^{-3}$ during these bonfire nights and contributing to ~60–70% of the C-PM$_1$.

### 3.4.3 Autumn PM$_{2.5}$ episodes

The PM$_{2.5}$ concentrations in Delhi ramp up during the autumn with some of the highest episodic concentrations observed during this period and often attributed to agricultural burning (Vijayakumar et al., 2016; Liu et al., 2018; Jethva et al., 2018). In 2017 the most polluted episodes were in the autumn with highest PM$_{2.5}$ hourly concentrations exceeding 500 μg m$^{-3}$ for

75–228 hours across the various locations in Delhi (DPCC monitoring stations and US Embassy). These autumn episodes constituted 80–100% (across sites) of the hours for which PM$_{2.5}$ exceeded 500 μg m$^{-3}$ in 2017 across Delhi. The highest PM concentrations within autumn were observed during the periods with relatively lower VC and when the wind was from the north or the northwest (Fig. 2). The concentrations were relatively lower for periods with higher VC values and when the wind was from the south. While some of these observations seem to support the role of agricultural burning in these episodic PM

concentrations, we plan to strengthen this hypothesis in a future study using composition data that we collect during the next autumn season.

### 3.5 Primary vs secondary

Positive matrix factorization (PMF) conducted on the ACSM mass spectra provided further information on the sources and atmospheric processes that affect NR-PM$_1$ concentrations in Delhi (Bhandari et al., 2018). The organic aerosol (OA) was

separated into two factors: primary OA (POA) and oxygenated OA (OOA), with periods when the POA factor further separated into hydrocarbon-like OA (HOA) and biomass-burning OA (BBOA). POA exhibited strong diurnal variability, reflecting the impact of primary combustion emissions modulated by diurnal cycles in the PBLH. The POA fraction of organics was generally highest during the nighttime (~50% for winter and ~40% for summer) and lowest during the daytime (~20% for both winter and summer). As observed in other megacities, OOA was the largest constituent of the organic aerosol throughout the year

(Jimenez et al., 2009), demonstrating the profound influence of secondary formation on particle concentrations in Delhi. OOA contributed to 50–80% of the organics almost year-round (Fig. S9). We estimated primary particulate matter (PPM = POA +



Chl + BC) and secondary particulate matter (SPM = OOA + $NH_4$ + $NO_3$ + $SO_4$) following Sun et al. (2013). Since chloride was considered primary and ammonium was generally highly correlated with chloride, we proportioned a chloride-equimolar amount of ammonium as primary and the remaining as secondary. In Fig. 11 we separate C-$PM_1$ into PPM and SPM by season and time of day. We observed that almost 50–70% of Delhi's C-$PM_1$ was secondary in nature for the winter and spring months and up to 60–80% for the warmer summer and monsoon months. Our results show that secondary aerosol accounts for the dominant fraction of Delhi's ambient NR-PM1 under most conditions. While our analyses do not provide direct evidence on the origin of the secondary fraction of $PM_1$, consideration of typical advection timescales from the upwind boundaries of Delhi (~2–3 h at typical wind speeds) suggests that a substantial fraction of Delhi's secondary aerosol may be transported from upwind regions, which also experience high PM mass loadings (Guttikunda and Goel, 2013). These findings suggest that improving Delhi's air quality will require a concerted effort at both at the local and the regional level. Future work could usefully apportion the composition of $PM_1$ at receptor sites upwind of Delhi.

BC was found to be well correlated ($R^2$ = 0.65) with HOA (for periods when HOA was a separate factor) (Bhandari et al., 2018), suggesting that traffic, diesel generators, and other liquid fossil fuel combustion contribute substantially to the BC inventory for Delhi. Furthermore, unlike chloride, the highest BC concentrations were uncorrelated with any particular wind direction (Fig. 8) and also showed less seasonal variation than other $PM_1$ species (Fig. 4) potentially indicating a nearby year-round source that was less affected by atmospheric mixing. We suspect that trucks (and other diesel vehicles) were a major source of the high BC concentrations that we observed, similar to what has been observed in other urban environments in India (Latha et al., 2004). While BC absolute concentrations did not vary as much as other $PM_1$ species, the fractional contribution to BC was as high as 20% during periods when the C-$PM_1$ was lower (Fig. 10). These findings indicate the large local nature of BC emissions and the potential to reduce BC concentrations by targeting high-emitters such as heavy-duty trucks and diesel generation systems (Baidya and Borken-Kleefeld, 2009). Previous studies have shown that a small fraction (10–20%) of high-emitting heavy-trucks contribute to almost half of the total BC emissions from heavy-duty trucks (Ban-Weiss et al., 2009).

## 4 Conclusions

We used continuous, highly time-resolved and long-term data to provide a detailed seasonal and diurnal characterization of Delhi's $PM_1$. We included data for organics, chloride, ammonium, nitrate, sulfate and BC from January-2017 to April-2018. The submicron mass for each species varied dynamically by season and by time of day. Meteorology was found to be an important factor in the modulation of PM levels, specifically by change in the VC that varied dynamically as the PBLH varied by season and time-of-day. The PM levels were generally the highest during the cooler months and times-of-day, periods when the VC values were the lowest. Furthermore, concentrations of volatile species (e.g., ammonium chloride) were further enhanced during the cooler periods when they had a higher tendency to be in the particle phase. While organics from biomass burning were enhanced during the cooler months, organics in general consistently (across seasons and times-of-day) contributed to ~60% of Delhi's $PM_1$. We observed some of the highest chloride concentrations measured anywhere in the world with average



concentrations higher than 50 µg m$^{-3}$ for periods during winter mornings when winds were from the northwest, resulting in part from what we suspect to be an industrial source.

We estimate that substantially more than half of Delhi's PM$_1$ is of secondary origin. In combination with other evidence, including the high levels of remotely-sensed PM$_{2.5}$ observed across the upwind states of Haryana and Punjab (Dey et al., 2012;
van Donkelaar et al., 2015), this finding points to the likely conclusion that the high pollution observed in Delhi is not merely a local problem, but one with a widespread regional source as well. Accordingly, reducing the PM levels in Delhi will require both a local and a regional effort with benefits that will be felt across the Indo-Gangetic plain. At the same time, primary PM$_1$ levels in Delhi are extremely high in absolute mass terms, and are likely driven principally by local emissions within the Delhi National Capital Region. Delhi's air pollution has many critical sources, some are local, some are regional (Chowdhury
et al., 2007; Guttikunda and Goel, 2013; Health Effects Institute, 2018). Coordinated regional and local controls of nearly all contributors will be required to bring about the order-of-magnitude concentrations reductions that will ultimately make the air safe to breathe (Kumar et al., 2013, 2015; Bhanarkar et al., 2018).

Long-term monitoring campaigns such as the DAS can contribute previously unavailable information on the evolving role of sources and other processes that govern air pollution in Indian cities. In particular, continuous, highly-time resolved data
provide a basis for evaluating the intended and unintended impact of policies and natural events on Delhi's air quality in near real-time. However, air pollution is spatially variable, and a single site generally does not provide sufficient information for the complete assessment of air quality in a large urban area like Delhi. Future work could usefully expand on this study through coordinated measurements of aerosol chemical composition at other locations. One key research need is to conduct similar measurements at sites upwind and downwind of Delhi to help more precisely quantify the role of local and regional sources in
driving Delhi's air pollution. Long-term studies of the changing nature of air pollution in South Asian cities can help inform much-needed efforts to protect a large part of the world's population from adverse effects of poor air quality.

## Appendix A

### A1  ACSM: calibration and operational procedures

Lens alignment and flow calibration were conducted at the start of the campaign. Ionizer tuning, quadrupole resolution ad-
justment, adjustment of multiplier voltage, and m/z calibration were conducted bimonthly. The pinhole was cleaned at least biweekly. Calibrations for the response factor (RF) of nitrate and the relative ionization efficiencies (RIEs) of ammonium and sulfate were conducted several times throughout the campaign (Table S2). For the RF and RIE calibrations, 300 nm particles, generated from 5mM solutions of ammonium nitrate and ammonium sulfate, were injected simultaneously into the ACSM and CPC. The size selected particles were sampled in jump mode (for all calibrations) as well as single scan mode (Sept 2017 and
Jan 2018), which is now the recommended procedure for this calibration. The RF/N$_2$ air beam ratio was consistent in all jump mode calibrations, suggesting a consistent sensitivity of the instrument. Thus, the RF and RIE values from the two single scan mode calibrations were used for all data (one value up to September-2017 and another value for the data post September-2017).



## A2   ACSM: data processing

Time dependent air beam corrections were applied to the raw data based on $N_2$ signal changes relative to the reference $N_2$ signal (when the calibration was performed). Relative Ion Transmission (RIT) correction was applied using the default RIT curve (not the measured RIT curve) because of the occurrence of a low naphthalene signal due to high concentration of m/z fragments

in sampling that build up and desorb during the filter sampling period (Aerodyne, personal communication). Detection limits were applied to species concentrations (Ng et al., 2011c) and data below the detection limit were replaced with 0.5 times the detection limit. CE was applied to account for inefficient aerosol collection due to effects such as particle bounce at the vaporizer. A composition dependent CE was calculated based on the method described in Middlebrook et al. (2012). An inline Nafion dryer lowered RH levels to less than 80% and ammonium nitrate fraction was less than 40% throughout the campaign.

Accordingly, only acidity dependent CE was applied. This method assumes that the particles are internally mixed and hence a single correction factor was applied for all species.

To account for particle loss during transmission through the aerodynamic lens, a transmission efficiency (TE) correction factor was computed using hourly averaged SMPS data. The following method was used to compute the TE correction factor:

Hourly particle density was computed using hourly averaged ACSM composition (DeCarlo et al., 2004). Ammonium was

attributed to each of the other inorganic species, assuming that ammonium would first neutralize sulfate, followed by nitrate and then chloride (Du et al., 2010). The tracer-based method was used to compute average organics composition and density (Ng et al., 2011b; Cross et al., 2007; Kuwata et al., 2012). Mobility diameter was converted to vacuum aerodynamic diameter ($D_{va}$) using the method described in DeCarlo et al. (2004), by assuming the Jayne shape factor to be 1 and calculated density. The averaged experimental TE curve of an aerodynamic focusing lens system (Liu et al., 2007) was applied to the particle size

distribution, and the TE correction factor was calculated as the ratio of total particle volume to the volume after applying the TE curve. Finally, average TE factors were computed for every hour of the day for every season (Fig. S10) and the ACSM concentrations were multiplied with this correction factor.

## Appendix B:  Inorganic modeling

The Extended Aerosol Thermodynamics Model (E-AIM) is used for interpreting the effect of gas-particle partitioning (GAP)

on the seasonality of concentrations (Friese and Ebel, 2010). The focus of this modeling is on inorganic species concentrations. While E-AIM can account for organic-inorganic interactions, since the identity of organic phase compounds is unknown, these interactions are ignored. Further, model IV of E-AIM is employed as it permits the variation of temperature and RH in the presence of the chloride anion. However, there are at least two limitations to the approach:

- The model always requires that charge balance be maintained, although charged aerosols have been previously reported
in literature. Further, the model does not provide a route to account for periods with excess cations; no additional anions are available in the model. $Na^+$, the only additional cation available and used as a surrogate for metal cations not measured in this study, is used to balance the charge for periods with excess anions.





- Periods with RH less than 60% cannot be run in the presence of chloride. To deal with this, RH for all such periods is set to 60%.

Due to data limitations and the above conditions, only periods between midnight–3 AM and 11 AM–midnight are analyzed. Hourly averaged diurnal NR-PM$_1$ species concentrations and gas phase NH$_3$ concentrations (obtained when available from the nearest central regulatory monitoring stations) for winter of 2017 are input into the model. This technique of running the model has been recently validated considering newly discovered issues in such thermodynamic models (Song et al., 2018; G. Murphy et al., 2017). The model is run in two modes—a "constrained" and an "unconstrained" mode. In the first run, diurnal data for winter of 2017 is input together with actual temperature and RH "constrained"; this mode forces the model to prevent gas-aerosol partitioning of the input data and instead generate equilibrium concentrations of gas-phase species HCl and HNO$_3$. Together with the measured NR-PM$_1$ speciated concentrations and NH$_3$, these concentrations are used to obtain total concentration estimates for NH$_3$ (NH$_3$+NH$_4^+$), NO$_3$ (HNO$_3$+NO$_3^-$), Cl (HCl+Cl$^-$) and H$^+$ (HCl+HNO$_3$). Other species are non-volatile, and their particle phase concentrations are their total concentrations. The obtained actual concentrations corrected for VC effects are run with the temperature and RH of summer 2017. Thus, to estimate maximum PM formation potential relative to the sources in winter 2017, diurnal "source" concentration averages for winter 2017 are applied to summer. We run the model in an "unconstrained" mode—the goal being to allow repartitioning for achieving equilibrium.

For winter of 2017, chloride and nitrate were almost completely in aerosol phase except between 12–5 PM (for analyzed periods >55% of chloride and >85% of nitrate in particle phase). Applying winter 2017 source strength to summer, we observe a significant shift—maximum nitrate in particle phase never exceeds 40% (10 µg m$^{-3}$) and chloride never exceeds 10% (3.5 µg m$^{-3}$). Thus, temperature and RH can explain the dramatic drop in concentrations of particle phase chloride and nitrate.

*Author contributions.* JSA, LHR, GH, SG and SB designed the study. SG, SB, PS, ZA and SS carried out the data collection. SG, SB, KP, SS, carried out the data processing and analysis. SG, SB, KP, DSW, LHR and JSA assisted with the interpretation of results. All co-authors contributed to writing and reviewing the manuscript.

*Competing interests.* The authors declare that they have no conflict of interest.

*Acknowledgements.* JSA was supported by the Climate Works Foundation. We are thankful to the Indian Institute of Technology Delhi (IITD) for institutional support. We are grateful to all student and staff members of the Aerosol Research Characterization laboratory (especially Nisar Baig and Yawar Hasan) and the Environmental Engineering laboratory (especially Sanjay Gupta) at IITD for their constant support. We are thankful to Philip Croteau (Aerodyne Research) and Maynard Havlicek (TSI) for always providing timely technical support for the instrumentation.



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





**Table 1.** Seasonal summary of PM$_1$ species—arithmetic mean (AM), geometric mean (GM), and geometric standard deviation (GSD) for hourly concentrations.

| | AM (µg m$^{-3}$) | | | | GM (µg m$^{-3}$) (GSD) | | | |
|---|---|---|---|---|---|---|---|---|
| | Winter | Spring | Summer | Monsoon | Winter | Spring | Summer | Monsoon |
| Org | 112 | 61 | 35 | 23 | 94 (1.9) | 49 (2.1) | 26 (2.1) | 18 (2.0) |
| NH$_4$ | 20 | 10 | 5.2 | 4.6 | 17 (2.0) | 6.4 (2.8) | 4.2 (1.9) | 3.8 (1.9) |
| Chl | 23 | 9.5 | 1.5 | 0.4 | 13 (3.2) | 2.9 (5.3) | 0.5 (3.5) | 0.3 (2.5) |
| NO$_3$ | 24 | 9.0 | 3.8 | 3.6 | 21 (1.7) | 5.9 (2.7) | 2.6 (2.4) | 2.4 (2.5) |
| SO$_4$ | 16 | 10 | 10 | 10 | 13 (1.9) | 8.7 (1.7) | 8.5 (1.8) | 8.1 (2.0) |
| BC | 15 | 11 | 9.0 | 11[a] | 10 (2.4) | 8.0 (2.3) | 6.3 (2.3) | 7.9 (2.4)[a] |
| NR-PM$_1$ | 195 | 100 | 55 | 41 | 168 (1.8) | 79 (2.1) | 45 (1.9) | 34 (1.9) |
| C-PM$_1$[b] | 209 | 110 | 64 | - | 182 (1.8) | 88 (2.1) | 52 (1.9) | - |
| SMPS-PM$_1$[c] | 199 | 115 | 78 | 57 | 166 (1.9) | 89 (2.1) | 60 (2.1) | 46 (1.9) |

[a]Based on limited BC data for monsoon due to instrument downtime.

[b]Composition-based estimate of PM$_1$ (BC + NR-PM$_1$)

[c]SMPS based estimate using hybrid PSD and assuming a density of 1.6 g cm$^{-3}$





**Table 2.** Day and night summary of $PM_1$ species and meteorological parameters. Arithmetic mean used for all species and parameters, except wind direction for which we used median to estimate its central tendency.

| | | Winter | | Spring | | Summer | | Monsoon | |
|---|---|---|---|---|---|---|---|---|---|
| | | Day | Night | Day | Night | Day | Night | Day | Night |
| Mass Concentration (μg m$^{-3}$) | Org | 86 | 138 | 47 | 76 | 29 | 41 | 20 | 26 |
| | Chl | 18 | 27 | 7.9 | 11 | 1.7 | 1.3 | 0.3 | 0.5 |
| | $NH_4$ | 19 | 21 | 9.3 | 11 | 5.5 | 4.8 | 4.9 | 4.4 |
| | $NO_3$ | 24 | 24 | 8.7 | 9.3 | 3.6 | 4.0 | 3.4 | 3.8 |
| | $SO_4$ | 16 | 15 | 9.8 | 10 | 10 | 9.5 | 11 | 9.1 |
| | BC | 9.4 | 20 | 6.8 | 15 | 5.2 | 13 | 5.4[a] | 18[a] |
| | NR-PM$_1$ | 163 | 226 | 82 | 117 | 50 | 61 | 39 | 44 |
| | C-PM$_1$[b] | 172 | 246 | 88 | 132 | 55 | 73 | - | - |
| | SMPS-PM$_1$[c] | 163 | 234 | 84 | 145 | 57 | 98 | 47 | 66 |
| | Temperature (°C) | 17 | 13 | 26 | 21 | 35 | 31 | 32 | 29 |
| | Relative Humidity (%) | 60 | 78 | 42 | 59 | 34 | 43 | 71 | 81 |
| Meteorological Parameters | Wind Speed (ms$^{-1}$) | 2.7 | 2.6 | 3.2 | 2.6 | 3.8 | 2.9 | 3.4 | 2.5 |
| | Wind Direction (°N) | 300 | 300 | 300 | 300 | 270 | 270 | 250 | 190 |
| | PBLH (m) | 920 | 340 | 1800 | 1000 | 2400 | 1600 | 1600 | 460 |

[a]Based on limited BC data for monsoon due to instrument downtime.

[b]Composition-based estimate of PM$_1$ (BC + NR-PM$_1$)

[c]SMPS based estimate using hybrid PSD and assuming a density of 1.6 g cm$^{-3}$



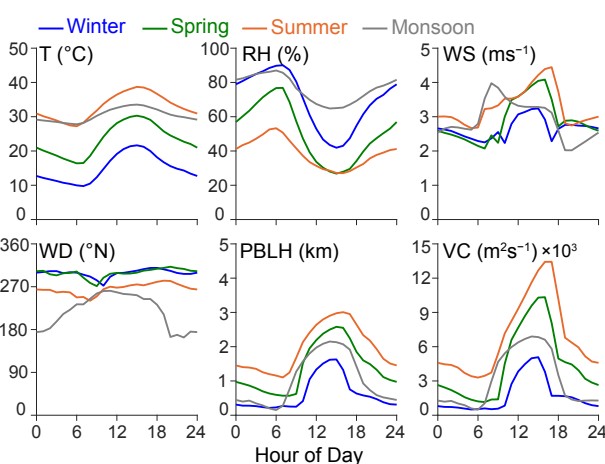

**Figure 1.** Diurnal profiles of meteorological parameters (temperature, relative humidity, wind speed, wind direction, PBLH and VC) by season. Average values by season and hour of day are presented for all parameters except wind direction. The median value is presented for wind direction. Ventilation coefficient (VC) = PBLH × wind speed.





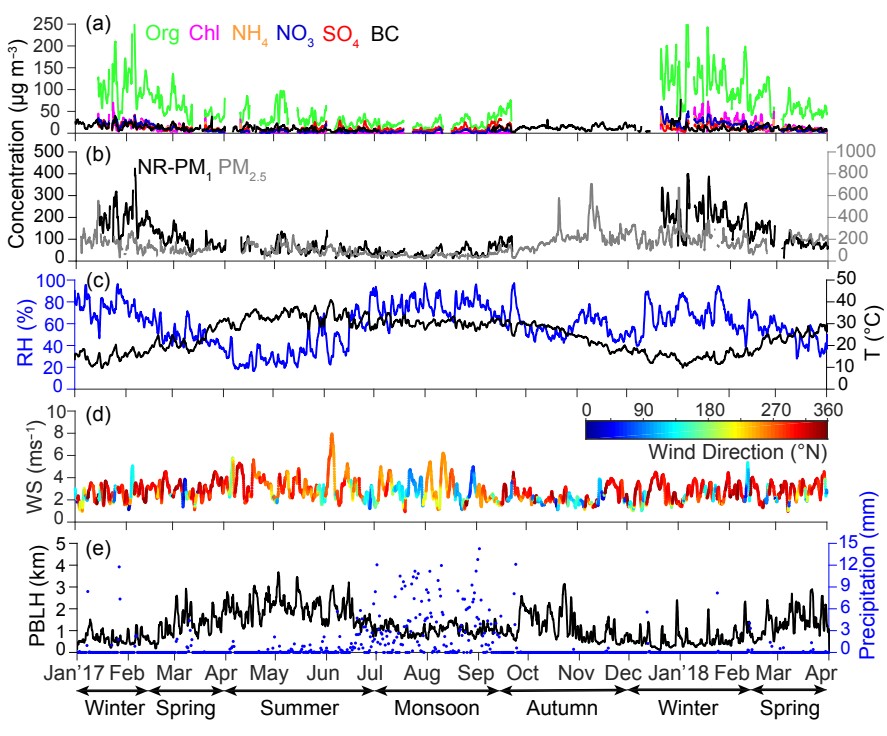

**Figure 2.** Time series of (a) $PM_1$ species (Org, Chl, $NH_4$, $NO_3$, $SO_4$, and BC), (b) NR-$PM_1$ and $PM_{2.5}$ (DPCC, R.K. Puram—3 km from our site), (c) relative humidity and temperature, (d) wind speed and direction, (e) PBLH and precipitation. A 24-hour moving average is applied on all time series presented.





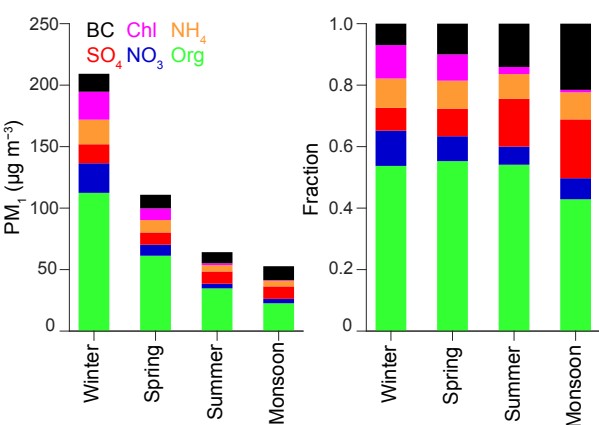

**Figure 3.** Average absolute and fractional composition of $PM_1$ (Org, Chl, $NH_4$, $NO_3$, $SO_4$, and BC) by season. Limited BC data for monsoon due to instrument downtime.





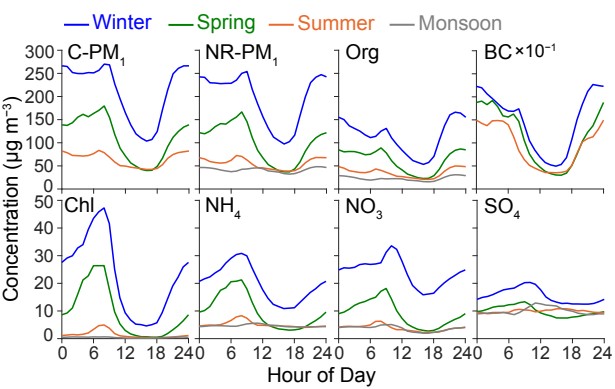

**Figure 4.** Average diurnal profiles of $PM_1$ species by season. Limited BC data for monsoon due to instrument downtime. Composition-based estimate of $PM_1$ (C-$PM_1$) = BC + NR-$PM_1$.





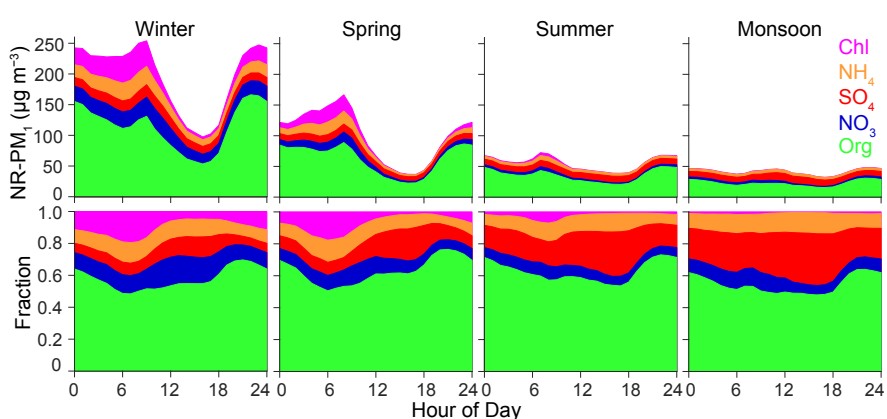

**Figure 5.** Stacked average absolute and fractional diurnal profiles of NR-PM$_1$ species by season.





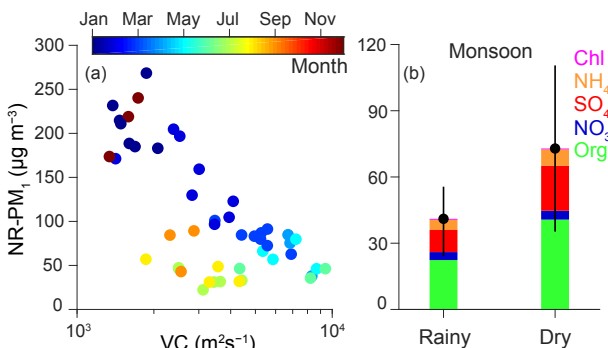

**Figure 6.** (a) Variations of NR-PM$_1$ mass concentrations as a function of ventilation coefficient. Each scatter point is a weekly average and is color-coded with the month. Note that July to mid-September is the monsoon season. (b) Average NR-PM$_1$ composition for days with (rainy) and without rain (dry). The vertical lines are the 25[th] (bottom) and 75[th] (top) percentiles.





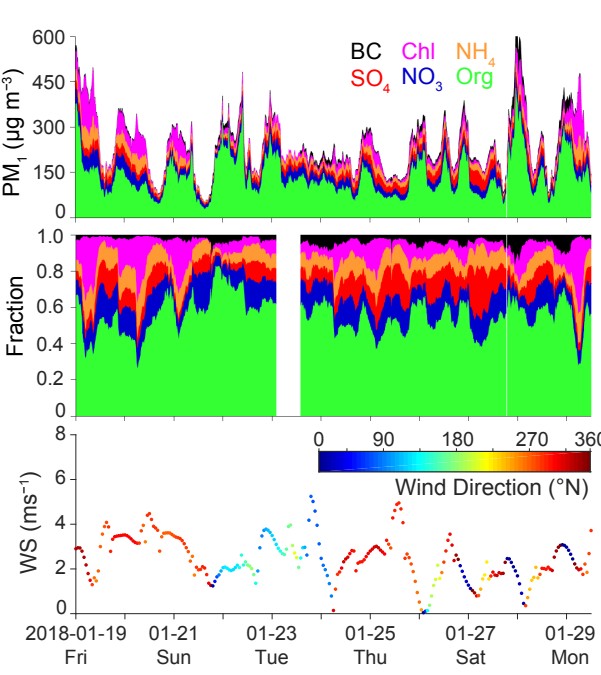

**Figure 7.** Time series of PM$_1$ species (Org, Chl, NH$_4$, NO$_3$, SO$_4$, and BC)—stacked absolute concentrations and fraction of PM$_1$—along with wind speed and wind direction for a period with high chloride concentrations.



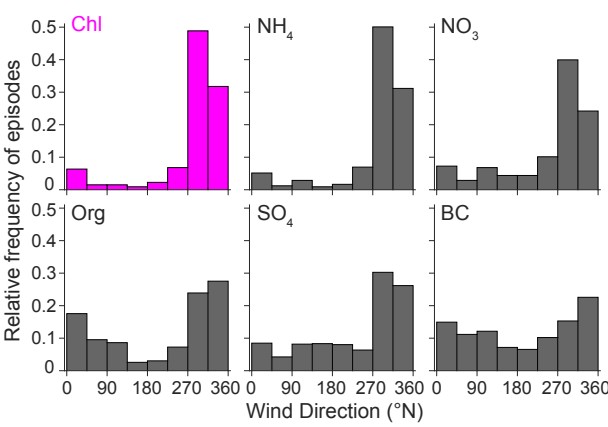

**Figure 8.** Probabilities for high episodic concentrations of PM$_1$ species (concentrations greater than the 90$^{th}$ percentile concentration of that species for the entire campaign) as a function of the wind direction.




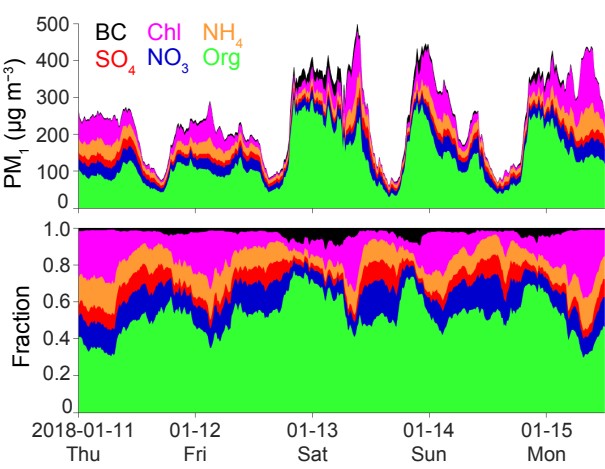

**Figure 9.** Time series of $PM_1$ species (Org, Chl, $NH_4$, $NO_3$, $SO_4$, and BC)—stacked absolute concentrations and fraction of $PM_1$—for a period with high organic PM concentration.



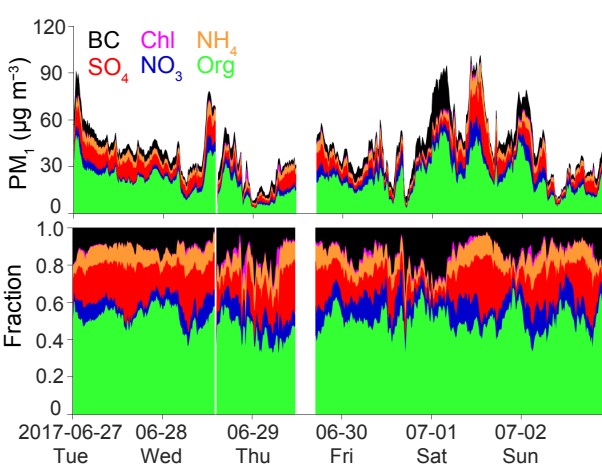

**Figure 10.** Time series of PM$_1$ species (Org, Chl, NH$_4$, NO$_3$, SO$_4$, and BC)—stacked absolute concentrations and fraction of PM$_1$—for a relatively less polluted (warm) period.




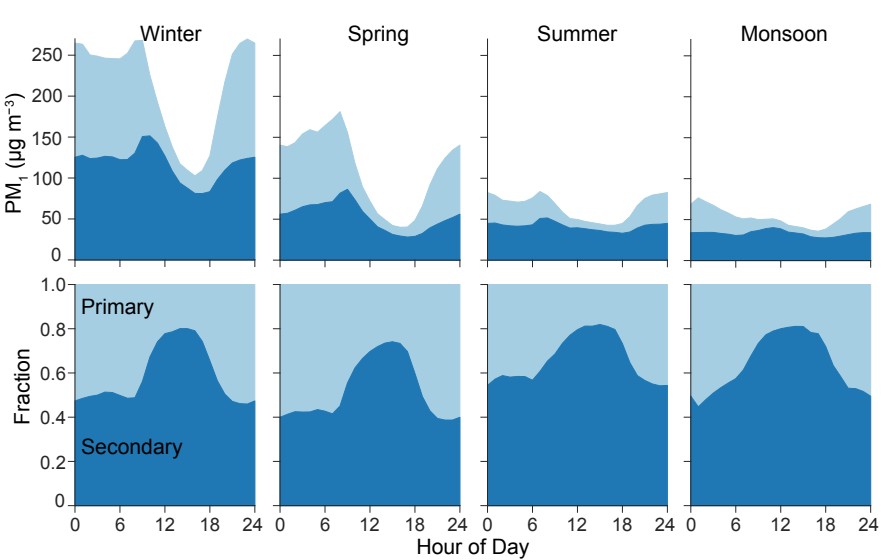

**Figure 11.** Average diurnal variation of mass concentrations and mass fractions of primary and secondary C-PM$_1$ by season.