# Peer review of "Submicron aerosol composition in the world's most polluted megacity: The Delhi Aerosol Supersite campaign"

_Atmospheric Chemistry and Physics, 2018_

## Referee Comment (RC1) · Anonymous Referee #1 · 29 Nov 2018

The study by Gani et al. reported submicron aerosol composition in a highly polluted city in India based on more than one-year measurement with an aerosol chemical speciation monitor. The seasonal variations in aerosol species, and roles of meteorology were characterized. Several episodes with extremely high concentrations of chloride and organics were discussed. The authors also investigated the relative importance of primary and secondary aerosol in different seasons. Overall, this study fits within the scope of ACP. Considering that Delhi, India can be a highly interested megacity in air pollution studies in the future, the results of this study are worth for publication. I have some comments below:

[Figure]

My major concern is the PMF analysis in this work. Although it seems not the focus of this study, the authors need to show more details about the PMF procedures, diagnostics, evaluation of the solutions. For instance, how PMF was performed? PMF was performed to the entire dataset or seasonally datasets? The authors mentioned that they can identify biomass burning OA factor during specific periods. I strongly encourage the authors to show more factors, which can help interpret the conclusions in the text (e.g., the impacts of agricultural burning in page 10, line 19).

The calculation of CE is a bit strange (page 13, line 10). The authors concluded that "the molar ratio of the inorganic anions to cations (ammonium) was 0.98 ($R^2$ = 0.89)" , suggesting that particle acidity should not be a factor affecting CE, while the authors applied acidity dependent CE. Suggest the authors presenting measured versus predicted ammonium. In addition, high concentration of chloride might not be completely in the form of ammonium chloride, may be KCl. This will also affect the estimation of particle acidity.

Page 13, line 9, the nafion dryer only decrease RH to less than 80%? Suppose to be much more efficient than this.

The PBLH in Figure 2 seems not right. For example, PBLH remained above 1.5 km from April to June. Please check the NASA meteorological reanalysis dataset. The PBLH output from MERRA2 might have large uncertainties.

---

## Referee Comment (RC2) · Anonymous Referee #2 · 29 Nov 2018

This manuscript presents analysis of PM1 (sub-micron) concentrations from a measurement campaign in Delhi, covering more than 1 year and multiple seasons including peak winter time episodes. Authors presented in this manuscript a detailed chemical analysis of the PM1 fraction using an online ACSM, investigated contributions of primary and secondary components in various seasons and for some specific episodes. Authors also presented an understanding of the role of meteorological conditions to observed shares of various chemical species. For Indian cities and especially for Delhi, this is a unique study. The methodology, location of interest (Delhi), analysis, and presentation of the results, fit the scope of ACP journal. I recommend this study for publication.

Minor comments:

PM1 is a subset of PM2.5 and the later is routinely monitored in Delhi. Section 3.1 discusses PM1 to PM2.5 fractions and concentrations are presented in Fig 2. Is it possible to add PM1 to PM2.5 ratio to the plot to see its variation over months? The lines are too close to decipher this.

An open ended conclusion in the study is source for chloride - is it biomass burning or an industrial source or both? Authors assumed that the region is ammonia rich - can this be corroborated with some ground measurements? Is the cluster for steel pickling large enough to support the peak concentrations?

Fig.11 suggests consistently high contribution of secondary PM fraction. While the PMF study (not included in this manuscript) might suggest more conclusions to its origins, given the information on wind speeds, wind directions, and secondary pm formation time scales, is it possible to hypothesize source regions (page 11, lines 5-10)?

---

## Referee Comment (RC3) · Anonymous Referee #3 · 20 Dec 2018

The authors present valuable data from the first long-term characterization of PM1 composition in Delhi. ACSM, SMPS, and aethalometer measurements are reported from January 2017 to April 2018. The paper is very well written and the data provide important insights into the sources of PM in Delhi. This work should be published in ACP after minor revisions.

- The ACSM was not functioning for part of the intense autumn pollution period (Oct 17-Jan 18), creating issues for the reporting of annual averages etc. Could the assumptions made to compensate for the missing data be checked or constrained using data from Autumn 2018?

- page 9 line 16 - delete 'perhaps'

- page 9 line 26 - I think it could be misleading to say that high levels of chloride observed don't exist elsewhere in S. Asia, given the data available. The measurements in other locations you cite here were not obtained with ACSM. Filter-based sampling can lead to revolatilization of PM components so it's possible the chloride existed but was not detected due to a sampling artifact. DeCarlo et al. detected high particulate chloride in Kathmandu using AMS. So, the discussion here should be framed differently - this is the first published report of high chloride, but it may actually be common across S. Asia, we don't know given the available data.

- The statements that more than half of Delhi's PM1 are secondary in origin are powerful and have a lot of significance (e.g. abstract, first full paragraph of page 12), but not enough information is given in this paper to support them. I guess the issue is that the PMF analysis has been saved for another paper which is not yet published. The paper probably would have been stronger if the two manuscripts were combined, but given the urgency and novelty of this data, I understand the decision to publish separately. I suggest that the discussion of the sources, except in cases where an obvious correlation between a source and the observations can be made (e.g. the Lohri fires discussion), should be removed or downplayed.

---

## Author Comment (AC1) · 8 Mar 2019

**Reviewer 1:**

**Comments:**

We would like to thank the reviewer for their comments.

The study by Gani et al. reported submicron aerosol composition in a highly polluted city in India based on more than one-year measurement with an aerosol chemical speciation monitor. The seasonal variations in aerosol species, and roles of meteorology were characterized. Several episodes with extremely high concentrations of chloride and organics were discussed. The authors also investigated the relative importance of primary and secondary aerosol in different seasons. Overall, this study fits within the scope of ACP. Considering that Delhi, India can be a highly interested megacity in air pollution studies in the future, the results of this study are worth for publication. I have some comments below:

My major concern is the PMF analysis in this work. Although it seems not the focus of this study, the authors need to show more details about the PMF procedures, diagnostics, evaluation of the solutions. For instance, how PMF was performed? PMF was performed to the entire dataset or seasonally datasets? The authors mentioned that they can identify biomass burning OA factor during specific periods. I strongly encourage the authors to show more factors, which can help interpret the conclusions in the text (e.g., the impacts of agricultural burning in page 10, line 19).

We agree with the reviewer. The details of the positive matrix factorization (PMF) are in Bhandari et al. (2019), which has now been submitted to ACP discussions for review (also included with our response to reviews). All of the points that the reviewer has noted have been discussed in detail in Bhandari et al. (2019).

The calculation of CE is a bit strange (page 13, line 10). The authors concluded that "the molar ratio of the inorganic anions to cations (ammonium) was 0.98 (R2 = 0.89)", suggesting that particle acidity should not be a factor affecting CE, while the authors applied acidity dependent CE. Suggest the authors presenting measured versus predicted ammonium. In addition, high concentration of chloride might not be completely in the form of ammonium chloride, may be KCl. This will also affect the estimation of particle acidity.

Thank you for bringing this point to our attention. We realized that there was a typo in the manuscript. It should say "the molar ratio of the inorganic anions to cations (ammonium) was 0.82 (R2 = 0.96)". The reviewer is correct in pointing to the potential role of cations other than ammonium. However, as discussed in Bhandari et al. (2019), the combined Organic-inorganic PMF results in a clear separation of the ammonium chloride factor when chloride concentrations are high.

Furthermore, in context of the CE correction, the sampling inlet has a Nafion membrane diffusion dryer that dries RH to less than 50% RH. Based on the Middlebrook et al. (2012) calculations of RH dependent collection efficiency (CE), CE is not affected by RH at RH<80%.

We have updated the text (Section 3.2):

"Over the campaign, the molar ratio of the inorganic anions to cations (ammonium) was 0.82 ($R2 = 0.96$)."

Page 13, line 9, the nafion dryer only decrease RH to less than 80%? Suppose to be much more efficient than this.

Yes, we did find that the nafion dryer was lowering the RH to within 50%. The 80% here was stated in context of CE correction (refer response to previous comment). The updated text (Appendix A2):

"A composition dependent CE was calculated based on the method described in Middlebrook et al. (2012). An inline Nafion dryer lowered RH levels to well below 50% (<80%) and the ammonium nitrate fraction was less than 40% throughout the campaign. Accordingly, we only applied the acidity dependent CE."

The PBLH in Figure 2 seems not right. For example, PBLH remained above 1.5 km from April to June. Please check the NASA meteorological reanalysis dataset. The PBLH output from MERRA2 might have large uncertainties.

We obtain the planetary boundary layer height (PBLH) data from NASA's meteorological reanalysis dataset (MERRA2). Our PBLH seasonal and diurnal averages (Figure 1) match those seen in literature (for e.g., Figure 6 from Tiwari et al. (2013)). In figure 2 we intend to present the long-term trends for various parameters, to which effect we use a 24 hr moving average to smoothen the timeseries. While the summertime (April to June) PBLH diurnal variation is between 100–3000 m, applying the moving average results in a reduced range. However, without applying an averaging filter, we cannot present anything meaningful in Figure 2.

References:

Bhandari, S., Gani, S., Patel, K., Wang, D. S., Soni, P., Arub, Z., Habib, G., Apte, J. S., and Hildebrandt Ruiz, L.: Sources and atmospheric dynamics of organic aerosol in New Delhi, India: Insights from receptor modeling, Submitted for peer review (Atmospheric Chemistry and Physics Discussions, acp-2019-230), 2019.

Middlebrook, A. M., Bahreini, R., Jimenez, J. L., and Canagaratna, M. R.: Evaluation of composition-dependent collection efficiencies for the Aerodyne aerosol mass spectrometer using field data, Aerosol Science and Technology, 46, 258–271, https://doi.org/10.1080/02786826.2011.620041, 2012.

Tiwari, S., Pandithurai, G., Attri, S., Srivastava, A., Soni, V., Bisht, D., Kumar, V. A., and Srivastava, M. K.: Aerosol optical proper- ties and their relationship with meteorological parameters during wintertime in Delhi, India, Atmospheric Research, 153, 465–479, https://doi.org/10.1016/j.atmosres.2014.10.003, 2015.

**Reviewer 2:**

We would like to thank the reviewer for their comments.

**Comments:**

This manuscript presents analysis of PM1 (sub-micron) concentrations from a measurement campaign in Delhi, covering more than 1 year and multiple seasons including peak winter time episodes. Authors presented in this manuscript a detailed chemical analysis of the PM1 fraction using an online ACSM, investigated contributions of primary and secondary components in various seasons and for some specific episodes. Authors also presented an understanding of the role of meteorological conditions to observed shares of various chemical species. For Indian cities and especially for Delhi, this is a unique study. The methodology, location of interest (Delhi), analysis, and presentation of the results, fit the scope of ACP journal. I recommend this study for publication.

Minor comments:

PM1 is a subset of PM2.5 and the later is routinely monitored in Delhi. Section 3.1 discusses PM1 to PM2.5 fractions and concentrations are presented in Fig 2. Is it possible to add PM1 to PM2.5 ratio to the plot to see its variation over months? The lines are too close to decipher this.

We agree with the limitation of only presenting PM1 and PM2.5, and not the ratio. However, unfortunately, we do not have PM2.5 measurements at our site and the ones we use in Fig. 2 are from a site 3 km (nearest site with available data). Furthermore, we often see instances when the PM1 at our site exceeds PM2.5 measured at other sites for which data is available — possibly higher influence of local sources at our site in combination with different measurement techniques. We think that presenting the ratio could potentially be misleading. That said, we will use scanning mobility particle sizer data from our site to discuss in detail how the particle size distribution changes over the seasons and times of day in a future publication.

An open ended conclusion in the study is source for chloride - is it biomass burning or an industrial source or both? Authors assumed that the region is ammonia rich - can this be corroborated with some ground measurements? Is the cluster for steel pickling large enough to support the peak concentrations?

Please note that we have included NH3 data from nearest regulatory agency monitor (Appendix A) and have cited satellite based NH3 estimates (Warner et al., 2017). The steel pickling cluster details are in Jaiprakash et al. (2017). Furthermore, the positive matrix factorization (PMF) results from Bhandari et al 2019 show that there is no meaningful correlation between chloride observed with biomass burning indicators.

Fig.11 suggests consistently high contribution of secondary PM fraction. While the PMF study (not included in this manuscript) might suggest more conclusions to its origins, given the information on wind speeds, wind directions, and secondary pm formation time scales, is it possible to hypothesize source regions (page 11, lines 5-10)?

We discuss these issues in detail in Bhandari et al. (2019) — the manuscript has been submitted for review to Atmospheric Chemistry and Physics Discussions (also included with our response to reviews).

References:

Bhandari, S., Gani, S., Patel, K., Wang, D. S., Soni, P., Arub, Z., Habib, G., Apte, J. S., and Hildebrandt Ruiz, L.: Sources and atmospheric dynamics of organic aerosol in New Delhi, India: Insights from receptor modeling, Submitted for peer review (Atmospheric Chemistry and Physics Discussions, acp-2019-230), 2019.

Warner, J. X., Dickerson, R. R., Wei, Z., Strow, L. L., Wang, Y., and Liang, Q.: Increased atmospheric ammonia over the world's major agricultural areas detected from space, Geophysical Research Letters, 44, 2875–2884, https://doi.org/10.1002/2016GL072305, 2017.

Jaiprakash, Singhai, A., Habib, G., Raman, R. S., and Gupta, T.: Chemical characterization of PM1.0 aerosol in Delhi and source apportionment using positive matrix factorization, Environmental Science and Pollution Research, 24, 445–462, https://doi.org/10.1007/s11356-016-7708-8, 2017.

**Reviewer 3:**

We would like to thank the reviewer for their comments.

**Comments:**

The authors present valuable data from the first long-term characterization of PM1 composition in Delhi. ACSM, SMPS, and aethalometer measurements are reported from January 2017 to April 2018. The paper is very well written and the data provide important insights into the sources of PM in Delhi. This work should be published in ACP after minor revisions.

- The ACSM was not functioning for part of the intense autumn pollution period (Oct 17-Jan 18), creating issues for the reporting of annual averages etc. Could the assumptions made to compensate for the missing data be checked or constrained using data from Autumn 2018?

This is a good point. In SI Table 1 we use various metrics to control for the missing data to the best of our ability. However, data from Autumn 2018 is beyond the scope of this manuscript.

- page 9 line 16 - delete 'perhaps'

We have corrected this in the revised manuscript.

- page 9 line 26 - I think it could be misleading to say that high levels of chloride observed don't exist elsewhere in S. Asia, given the data available. The measurements in other locations you cite here were not obtained with ACSM. Filter-based sampling can lead to revolatilization of PM components so it's possible the chloride existed but was not detected due to a sampling artifact. DeCarlo et al. detected high particulate chloride in Kathmandu using AMS. So, the discussion here should be framed differently - this is the first published report of high chloride, but it may actually be common across S. Asia, we don't know given the available data.

Yes, this is a good point and we have rephrased in the revised manuscript to make our findings clearer in the context of other studies from South Asia using similar online instrumentation. However, we should note that the concentrations we observe in Delhi are at least an order of magnitude higher than those detected by groups in South Asia using online aerosol mass spectrometers, including DeCarlo and colleagues in Kathmandu (Nepal) and by those outside of Delhi in North India (Goetz et al., 2018; Chakraborty et al., 2015).

We have added the following text (Section 3.4.1):

"While filter based studies can cause underreporting of volatile species such as ammonium chloride, the levels of chloride we observe in Delhi are much higher than those reported from studies in South Asia (outside Delhi) that use online aerosol instrumentation (Goetz et al., 2018; Chakraborty et al., 2015)."

- The statements that more than half of Delhi's PM1 are secondary in origin are powerful and have a lot of significance (e.g. abstract, first full paragraph of page 12), but not enough

information is given in this paper to support them. I guess the issue is that the PMF analysis has been saved for another paper which is not yet published. The paper probably would have been stronger if the two manuscripts were combined, but given the urgency and novelty of this data, I understand the decision to publish separately. I suggest that the discussion of the sources, except in cases where an obvious correlation between a source and the observations can be made (e.g. the Lohri fires discussion), should be removed or downplayed.

We fully agree with the spirit of this comment. All of the points made have been addressed in Bhandari et al. (2019), which has now been submitted for peer review to Atmospheric Chemistry and Physics Discussions (also included with our response to reviews).

References:

Goetz, J. D., Giordano, M. R., Stockwell, C. E., Christian, T. J., Maharjan, R., Adhikari, S., Bhave, P. V., Praveen, P. S., Panday, A. K., Jayarathne, T., Stone, E. A., Yokelson, R. J., and DeCarlo, P. F.: Speciated on-line PM1 from South Asian combustion sources: Part I, Fuel- based emission factors and size distributions, Atmospheric Chemistry and Physics Discussions, 2018, 1–37, https://doi.org/10.5194/acp- 2018-369, 2018.

Chakraborty, A., Bhattu, D., Gupta, T., Tripathi, S. N., and Canagaratna, M. R.: Real-time measurements of ambient aerosols in a polluted Indian city: Sources, characteristics, and processing of organic aerosols during foggy and nonfoggy periods, Journal of Geophysical Research: Atmospheres, 120, 9006–9019, https://doi.org/10.1002/2015JD023419, 2015.

Bhandari, S., Gani, S., Patel, K., Soni, P., Wang, D. S., Arub, Z., Habib, G., Apte, J. S., and Hildebrandt Ruiz, L.: Sources and atmospheric dynamics of organic aerosol in New Delhi, India: Insights from receptor modeling, Submitted for peer review (Atmospheric Chemistry and Physics Discussions, acp-2019-230), 2019.